# Design, Synthesis and Characterization of HIV-1 CA-Targeting Small Molecules: Conformational Restriction of PF74

**DOI:** 10.3390/v13030479

**Published:** 2021-03-15

**Authors:** Rajkumar Lalji Sahani, Raquel Diana-Rivero, Sanjeev Kumar V. Vernekar, Lei Wang, Haijuan Du, Huanchun Zhang, Andres Emanuelli Castaner, Mary C. Casey, Karen A. Kirby, Philip R. Tedbury, Jiashu Xie, Stefan G. Sarafianos, Zhengqiang Wang

**Affiliations:** 1Center for Drug Design, College of Pharmacy, University of Minnesota, Minneapolis, MN 55455, USA; rsahani@umn.edu (R.L.S.); raquel.diana.rivero@gmail.com (R.D.-R.); vvsanjeev99@gmail.com (S.K.V.V.); leiwang@dlut.edu.cn (L.W.); jxie@umn.edu (J.X.); 2Laboratory of Biochemical Pharmacology, Department of Pediatrics, Emory University School of Medicine, Atlanta, GA 30322, USA; haijuan.du@emory.edu (H.D.); huanchun.zhang@emory.edu (H.Z.); andres.emanuelli@emory.edu (A.E.C.); karen.kirby@emory.edu (K.A.K.); philip.tedbury@emory.edu (P.R.T.); stefanos.sarafianos@emory.edu (S.G.S.); 3Children’s Healthcare of Atlanta, Atlanta, GA 30322, USA; 4Department of Molecular Microbiology and Immunology, School of Medicine, Christopher S. Bond Life Sciences Center, University of Missouri, Columbia, MO 65211, USA; mcc6x2@mail.missouri.edu

**Keywords:** HIV-1, capsid protein, PF74, conformational constraint, metabolic stability

## Abstract

Small molecules targeting the PF74 binding site of the HIV-1 capsid protein (CA) confer potent and mechanistically unique antiviral activities. Structural modifications of PF74 could further the understanding of ligand binding modes, diversify ligand chemical classes, and allow identification of new variants with balanced antiviral activity and metabolic stability. In the current work, we designed and synthesized three series of PF74-like analogs featuring conformational constraints at the aniline terminus or the phenylalanine carboxamide moiety, and characterized them using a biophysical thermal shift assay (TSA), cell-based antiviral and cytotoxicity assays, and in vitro metabolic stability assays in human and mouse liver microsomes. These studies showed that the two series with the phenylalanine carboxamide moiety replaced by a pyridine or imidazole ring can provide viable hits. Subsequent SAR identified an improved analog **15** which effectively inhibited HIV-1 (EC_50_ = 0.31 μM), strongly stabilized CA hexamer (ΔTm = 8.7 °C), and exhibited substantially enhanced metabolic stability (t_1/2_ = 27 min for 15 vs. 0.7 min for PF74). Metabolic profiles from the microsomal stability assay also indicate that blocking the C5 position of the indole ring could lead to increased resistance to oxidative metabolism.

## 1. Introduction

The capsid protein (CA) of human immunodeficiency virus type 1 (HIV-1), expressed as part of the viral Gag polyprotein, plays a critical role in multiple steps of the viral replication cycle [1]. In the late stage, CA–CA interactions drive the assembly of Gag polyproteins, and the reassembly of released CAs, toward immature and mature viral capsids, respectively [2]. In the early stage, productive HIV-1 infection entails numerous CA-mediated post-entry events [1] up to the completion of integration, including uncoating, cytoplasmic trafficking, reverse transcription, nuclear import, and integration site targeting. These events typically involve CA binding by host proteins, such as nucleoporin 153 (NUP153) [3] and cleavage and polyadenylation specificity factor 6 (CPSF6) [4,5], both of which are required for the nuclear import of HIV-1 pre-integration complexes (PICs). CPSF6 is also required for viral integration site targeting [6,7]. In addition, the CA-cyclophilin A (CypA) interaction allows HIV-1 to evade restriction by host factor tripartite motif-containing protein 5 alpha (TRIM5a) [8]. Therefore, CA-targeting small molecules could inhibit viral replication via two distinct mechanisms of action: by disrupting CA–CA interactions, they could alter the overall core stability and impact viral assembly and uncoating; by competing against NUP153 and CPSF6 for CA binding, they could block viral nuclear entry and/or productive integration. These mechanisms, along with CA as an antiviral target, have been strongly validated with a few chemically distinct CA-targeting small molecule chemotypes [9], including mainly BI-2 [10], BM-4, BD-3 [11], CAP-1 [12], PF74 [13], GS-6207 (lenacapavir) [14,15]. Structurally, the CA monomer contains two primarily helical domains: the N-terminal domain (CA_NTD_) comprised of seven alpha-helices H1-7, and the C-terminal domain (CA_CTD_) consisted of four alpha-helices H8-11 [16,17]. Reported CA-binding compounds predominantly target the CA_NTD_ where multiple binding sites have been identified. Amongst these, the PF74 binding pocket [17] is particularly well-characterized and significant. PF74 binds to the interface formed between the H3 and H4 of CA_NTD_ and the H8 and H9 of the adjacent CA_CTD_ within a CA hexamer (Figure 1A) [13,17]. In this binding pocket, PF74 interacts with N57 of H3 through the carbonyl-O4 and N(2)-H of the phenylalanine moiety (Figure 1C), K70 and Q63 of H4 through the other carbonyl-O3 and the indole N(1)-H, and several other residues, N53, M66, A105, T107, and Y130, though hydrophobic interactions (Figure 1A). Two other CA-targeting chemotypes, BI-2 [10] and GS-6207 [14,15], also bind to the same pocket, with very similar backbone binding (Figure 1B), despite their hugely different structural complexities (Figure 1C). Furthermore, the PF74 binding site also accommodates CPSF6 and NUP153, the host factors required for nuclear entry and integration site targeting of viral PICs (Figure 1D) [18]. Therefore, small molecules targeting this binding pocket can interfere with both early and late events of viral replication, and confer unique antiviral profiles with a bimodal mechanism of action, as manifested with PF74 and GS-6207 [19].

Although GS-6207 is far superior in antiviral potency to PF74, it lacks sufficient aqueous solubility to be an oral drug. Furthermore, mutations conferring resistance to GS-6207 have been selected in vitro [14,15]. One of the mutations that appeared in vitro, Q67H, was reported in recent clinical trials to be present in a patient sample, after nine days of treatment with GS-6207 [14]. On the other hand, PF74 suffers from extremely low metabolic stability [20,21,22,23]. These deficiencies necessitate continued medicinal chemistry efforts in identifying additional chemotypes targeting the PF74 pocket. For this purpose, the structural simplicity of PF74 offers a distinct synthetic chemistry advantage over GS-6207 for chemical modifications. We have previously pursued the design and synthesis of PF74-like compounds and have identified several subtypes with improved antiviral potency and/or enhanced metabolic stability [20,21,22,23]. The current work studies the impact of overall molecular rigidity on CA targeting and metabolic stability. The primary rationale is that binding to a protein typically results in the conformational/configurational restriction of a ligand, and hence, an entropy loss that needs to be compensated for, the amount of which will be reduced if the ligand is conformationally restricted to begin with [24]. The importance of rigidity in small molecule inhibitor design has been demonstrated [25,26]. In addition, PF74 features an aniline *N*-methyl group, which is a well-known metabolic handle [27,28] and a likely culprit of the poor metabolic stability of PF74. In our redesign of PF74 (Figure 2), rigidity is introduced either to the aniline moiety by capping the *N*-methyl and the ortho–phenyl end of PF74 (modification A, chemotype 1), or to the phenylalanine amide moiety by capping the *N*-methyl and the amide carbonyl (modification B, chemotype 2 and chemotype 3). The subsequent structure–activity relationship (SAR) also explores previously reported [20,22] key interactions, such as halogen bonding conferred by the para-halogen (Figure 2, R^1^) of the aniline phenyl to N74, hydrophobic interaction by the indole N-Et (R^4^), and H-bonding by the indole C5-OH (R^3^) to K182 of the adjacent CA_CTD_.

## 2. Materials and Methods

### 2.1. Chemistry

All tested analogs were synthesized as described in Scheme 1, Scheme 2 and Scheme 3, and were fully characterized with ^1^H and ^13^C NMR, and HRMS. Synthetic procedures and compound characterization data are included in Appendix A. The general synthetic strategies for major analogs (**1**–**25**) tested in this work are described here (Scheme 1, Scheme 2 and Scheme 3) and the synthesis of other intermediates is outlined in Appendix A.

Chemotype 1 was synthesized according to procedures describe in Scheme 1 [20]. Commercially available indole-acetic acids **26** were treated with commercially available *L*-Phenylalanine methyl ester hydrochloride under a well-established strategy using HATU in the presence of DIPEA to afford **27**. LiOH hydrolysis of intermediates **27** afforded acid intermediates **28**, which were further reacted with corresponding indoline or tetrahydroquinoline to give the desired compounds **1**–**3**.

The synthesis of intermediates **30**–**37** and major analogs (**4**–**19**) was achieved using a route described in Scheme 2 [29,30]. Commercially available 3-bromopicolinic acid **29** was converted to 3-bromopicolinaldehyde **30** through oxalyl chloride/DMF assisted esterification, followed by DIBAL-H reduction. Intermediate **30** was then treated with commercially available (*R*)-(+)-2-Methyl-2-propanesulfinamide to form sulfonimine **31**, which was subjected to Grignard addition of benzyl-magnesium chloride to produce two diastereomers **32** and **33** in 3:1 ratio. Subsequently, intermediate **32** was subjected to Suzuki coupling with various boronic acids to produce intermediates **34**, which upon deprotection furnished intermediates **35**. Finally, intermediates **35** were reacted with corresponding acids with HATU in the presence of DIPEA to afford compounds **4**–**8**. Similarly, the synthesis of analogs **9**–**19** was accomplished from intermediate **33**, with the exception of using T_3_P in place of HATU in the final amide-coupling reaction for the synthesis of compound **15**.

Analogs **20**–**25** were synthesized as described in Scheme 3 [30]. The synthesis of intermediates **39**–**41** is discussed in detail in the Appendix A. Commercially available N-boc phenyl-alaninal **38** and glyoxal were treated with ammonia to construct imidazole ring intermediate **39**, which was subjected to copper catalyzed Chan-Lam reaction with various boronic acids to produce intermediate **40**. TFA deprotection of Boc group resulted in intermediate **41**, which was reacted with various indole-acetic acids using HATU in the presence of DIPEA to afford compounds **20**–**25**.

### 2.2. Cells

TZM-GFP cells are a modified version of TZM-bl cells and contain an integrated nlsGFP reporter gene under the transcriptional control of the HIV-1 long terminal repeat (LTR) [31,32]; TZM-GFP cells produce GFP following infection with HIV-1. TZM-GFP cells were kindly provided by Dr. Marc Johnson (University of Missouri-Columbia, Columbia, MO, USA) and cultured in DMEM supplemented with 10% fetal bovine serum (FBS; Hyclone, Logan, UT, USA). HEK293-FT cells were cultured in DMEM supplemented with 10% FBS. MT-2 cells were grown in RPMI supplemented with 10% heat-inactivated FBS. All cells were grown and maintained in humidified atmosphere containing 5% CO_2_ at 37 °C.

### 2.3. Method Details

#### 2.3.1. Thermal Shift Assays (TSAs) to Screen Compounds for Effect on HIV-1 CA Hexamer Stability

Compounds were screened for binding using purified covalently-crosslinked hexameric CA^A14C/E45C/W184A/M185A^ (CA121). CA121 cloned in a pET11a expression plasmid was kindly provided by Dr. Owen Pornillos (University of Virginia, Charlottesville, VA, USA). Protein was expressed in *E. coli* BL21(DE3)RIL and purified as reported previously [33]. The TSA has been previously described [20,21,22,23]. Briefly, the TSA was conducted on the PikoReal Real-Time PCR System (Thermo Fisher Scientific, Waltham, MA, USA) or the QuantStudio 3 Real-Time PCR system (Thermo Fisher Scientific, Waltham MA, USA). Each reaction contained 7.5 µM final concentration CA121, 1× Sypro Orange Protein Gel Stain (Life Technologies, Carlsbad, CA, USA) in 50 mM sodium phosphate buffer (pH 8.0) and 1% DMSO (control) or 20 µM compound in 1% DMSO. The plate was heated from 25 to 95 °C with a heating rate of 0.2 °C/10 s. The fluorescence intensity was measured with an Ex range of 475–500 nm and Em range of 520–590 nm. The differences in the melting temperature (ΔTm) of CA hexamer in DMSO (T_0_) verses in the presence of compound (Tm) were calculated using the following formula: ΔTm = Tm − T_0_.

#### 2.3.2. Virus Production

The wild-type laboratory HIV-1 strain, HIV-1_NL4-3_ [34], was produced using a pNL4-3 vector that was obtained through the NIH AIDS Reagent Program, Division of AIDS, NIAID, NIH. HIV-1NL4-3 was generated by transfecting HEK 293FT cells in a 10 cm tissue culture dish with 10 µg of the pNL4-3 vector and X-tremeGENE HP DNA Transfection Reagent (Roche, Indianapolis, IN, USA), following the manufacturer’s protocol. Supernatant was harvested 48–72 h post-transfection. The viral supernatant was then concentrated using Lenti-X Concentrator (Takara, Mountain View, CA, USA), following the manufacturer’s protocol. The resulting viral-containing pellet was concentrated 10-fold by resuspension in DMEM without FBS and stored at −80 °C.

#### 2.3.3. Anti-HIV-1 and Cytotoxicity Assays

Anti-HIV-1 activity of PF74 and PF74-related analogs were examined in TZM-GFP cells. The potency of HIV-1 inhibition by a compound was based on its inhibition of HIV-1 infection of TZM-GFP cells compared with that of compound-free (DMSO) controls. Briefly, TZM-GFP cells were plated at a density of 1 × 10^4^ cells per well in a 96-well plate. 24 h later, medium was replaced with medium containing compound dilutions. 24 h post treatment, cells were exposed to HIV-1_NL4-3_ (MOI = 0.1). After incubation for 48 h, anti-HIV-1 activity was assessed by counting the number of GFP positive cells on a Cytation 5 Imaging Reader (BioTek, Winooski, VT, USA) and 50% effective concentration (EC_50_) values were determined.

Cytotoxicity of each compound was also determined in TZM-GFP cells. Cells were plated at a density of 1 × 10^4^ cells per well in a 96-well plate and were continuously exposed to medium containing compound dilutions for 72 h. Cell viability was determined using a Cell Proliferation Kit II (XTT), and 50% cytotoxicity concentration (CC_50_) values were determined. All the cell-based assays were conducted with technical duplicates and with at least two independent experiments and the average values were determined.

For the EC_50_ and CC_50_ dose responses, values were plotted in GraphPad Prism 5 and analyzed with the log (inhibitor) vs. normalized response–variable slope equation. Final values were calculated in each independent assay and the average values were determined. Statistical analysis (calculation of standard deviation) was performed using Microsoft Excel.

### 2.4. Metabolic Stability Assay

Microsomal stability. The in vitro microsomal stability assay was conducted in triplicate in mouse and human liver microsomal systems, which were supplemented with nicotinamide adenine dinucleotide phosphate (NADPH) as a cofactor. Briefly, a compound (1 µM final concentration) was spiked into the reaction mixture containing liver microsomal protein (0.5 mg/mL final concentration) and MgCl_2_ (1 mM final concentration) in 0.1 M potassium phosphate buffer (pH 7.4). The reaction was initiated by addition of 1 mM NADPH, followed by incubation at 37 °C. A negative control was performed in parallel without NADPH to reveal any chemical instability or non-NADPH dependent enzymatic degradation for each compound. At various time points (0, 5, 15, 30 and 60 min), 1 volume of reaction aliquot was taken and quenched with 3 volumes of acetonitrile containing 0.1% formic acid. The samples were then vortexed and centrifuged at 15,000 rpm for 5 min at 4 °C. The supernatants were collected and analyzed by LC/MS/MS to determine the remaining percentage and in vitro metabolic half-life (t_1/2_).

### 2.5. Molecular Modeling

Molecular modeling was performed using the Schrödinger small molecule drug discovery suite 2019-4 [35]. The crystal structure of PF74 [17] in complex with native HIV-1 capsid protein was retrieved from the protein data bank (PDB code: 4XFZ [17]). Selected structures were evaluated for their interaction with native HIV-1 capsid protein using Maestro [36] (Schrödinger; LLC: New York, NY, USA) by subjecting to a docking protocol, involving preparing protein of interest, grid generation, ligand preparation, and docking. Post processing of each docked pose was done by PyMOL [37] (Schrödinger; LLC: New York, NY, USA). The native HIV-1 capsid protein/PF74 crystal structure was refined using the protein preparation wizard [38] (Schrödinger; LLC: New York, NY, USA) in which missing hydrogen atoms, side chains, and loops were added using Prime; waters beyond 5 Å were deleted, and minimized using the OPLS3e force field [39] to optimize the hydrogen bonding network and converge the heavy atoms to an RMSD of 0.3 Å. The active site around the native ligand PF74 was defined by the receptor grid generation tool in Maestro (Schrödinger; LLC: New York, NY, USA) covering all the residues within 12 Å. All the compounds were drawn using 2D-sketcher in Maestro and subjected to LigPrep to generate conformers and possible protonation at pH of 7 ± 2. All the dockings were performed using Glide XP [40] (Glide, version 8.2) with the van der Waals radii of nonpolar atoms for each of the ligands scaled by a factor of 0.8 to decrease penalties for close contacts. All docked poses were subjected to post-docking minimization (PMD) to minimize a small number of poses within the field of the receptor to produce better poses. The residue numbers of HIV-1 capsid protein used in the discussion and the figures were based on the native HIV-1 capsid protein.

## 3. Results

Analogs synthesized for each series were first tested in a biophysical thermal shift assay (TSA), which measures how compounds affect the stability of covalently crosslinked CA hexamers. TSA results are presented as the change of protein melting temperature in the presence of a compound compared to the protein melting temperature in the presence of DMSO control (ΔTm). A right shift (positive ΔTm) denotes a stabilizing effect, whereas a left shift (negative ΔTm) indicates a destabilizing effect. All final compounds were also tested in cell-based assays to determine anti-HIV-1 activity and cytotoxicity. The initial antiviral screening was conducted at 20 µM, from which compounds showing significant inhibition were further tested in a dose-response assay to determine antiviral EC_50_ values. CC_50_ values were determined by cytotoxicity assay. The parent PF74 was used as a control in these assays (ΔTm = 6.9 °C, EC_50_ = 0.70 μM, CC_50_ = 76 μM). Selected compounds were also tested for metabolic stability in liver microsomes. Molecular modeling was performed for a few analogs to corroborate the binding modes and understand the SAR.

### 3.1. The Indoline and Quinoline Analogs (Chemotype 1)

This short series sought to explore the effect of rigidifying the terminal *N*-methyl aniline. As shown in Table 1, cyclization of the *N*-methyl to the ortho position of the phenyl ring via a five-membered or six-membered ring is not tolerated as the resulting analogs **1**–**3** showed no significant antiviral activity or CA-stabilizing/destabilizing effect. This is likely to be due to the loss of an important van der Waals interaction of the *N*-methyl group with the side chain of N53 [20].

### 3.2. SAR of the Pyridine Series (Chemotype 2)

This chemotype features a key backbone conformational constraint conceived via building the phenylalanine carboxamide into a pyridine ring (Figure 2, modification B). Incidentally, such a design resulted in a phenylalanine mimic similar to the core of GS-6207 (Figure 1C). The prototype of this chemotype, analog **9** (Table 2), considerably inhibited HIV-1 (EC_50_ = 2.6 μM) and strongly stabilized CA hexamer (ΔTm = 4.5 °C). In stark contrast, the enantiomer of **9**, the *R*-phenylalanine derived analog **4**, was completely inactive in both the antiviral assay and the TSA (Table 2). This dramatic SAR trend was largely retained as additional *R*-configured analogs (**5**–**8**) showed neither CA-stabilizing nor destabilizing effect, and they lacked significant antiviral activity, whereas *S*-configured analogs **10**–**19** all inhibited HIV-1 at low to sub-micromolar concentrations (Table 2). The stereochemical requirement of the *S* configuration (Figure 2) is not unexpected since the phenyl ring of the phenylalanine serves as an anchor for ligand and host factor binding [41]. Among the *S*-configured analogs, the ones with an indazole (**10**) or a pyrazole (**11**) in place of an indole only weakly stabilized CA hexamers, and both inhibited HIV-1 at low micromolar EC_50_ concentrations. By contrast, with the exception of **12**, all indole analogs (**13**–**19**) strongly stabilized CA-hexamers (ΔTm = 5.9–8.7 °C). Notably, having a chlorine atom at the para position of the terminal phenyl ring (**13**–**15**) appeared to benefit both the antiviral activity and the CA-stabilizing effect, which may reflect halogen bonding with the CA-protein [22,23]. The OH group at the indole C5 position also conferred improved potency and CA-stabilizing effect (**15** vs. **13**), though the OH effect was not as prominent as observed with PF74 analogs [22]. In the latter case, OH is believed to form an H-bond with the K182 of the adjacent CA_CTD_. When the indole N is ethylated, the resulting analogs were considerably less active (**16** vs. **13**, **17** vs. **14**), which amounts to a reversed SAR trend compared to PF74 analogs [20]. Interestingly, significant difference was not observed between Br and Cl (**19** vs. **16**, **18** vs. **17**). Finally, compounds with a halogen (**13**–**19**) of this series were all somewhat cytotoxic. Overall, **15** was the most potent compound of this series (EC_50_ = 0.31 μM, Table 2 and Figure 3), with a better selectivity index (SI) over the parent PF74 (SI = 142 for **15** vs. 109 for PF74). A Wilcoxon rank-sum test was performed to determine the significance of the EC_50_ value of compound **15** compared to the EC_50_ value of the parent PF74 compound (EC_50_ = 0.7 μM). The *p*-value = 0.029, which demonstrates that difference between these two samples is statistically significant.

### 3.3. SAR of the Imidazole Series (Chemotype 3)

This series was designed by converting the phenylalanine carboxamide moiety into an imidazole ring for achieving backbone conformational constraint. Similar to the pyridine series (chemotype 2), the PF74-derived prototypical imidazole analog **20** displayed significant antiviral activity (EC_50_ = 3.7 μM) and strong CA-stabilizing effect (ΔTm = 3.9 °C). However, subsequent SAR did not yield improved compounds, with analogs **21**, **23**–**25** all showing similar activity profiles (Table 3). The single exception is compound **22** featuring a para-tetrazole group on the terminal phenyl ring, which conferred no antiviral activity. Interestingly, halogenated analogs within this series (**23**–**25**) were substantially less cytotoxic than the pyridine series (**13**–**19**).

### 3.4. Metabolic Stability

A major deficiency of PF74 as an antiviral compound is the extremely low metabolic stability [42,43,44], with half-life (t_1/2_) less than 1 min in both human liver microsomes (HLMs) and mouse liver microsomes (MLMs) [20,21,22,23]. Our previous work also revealed that this is mainly due to the poor resistance to cytochrome P450 3A4 (CYP3A4)-mediated oxidative metabolism, as the half-life was drastically elongated in the presence of a CYP3A4 inhibitor [20,21,22,23]. To assess the impact of conformational restriction on the susceptibility toward CYP3A4, we tested seven selected analogs in the microsomal stability assay. Overall, very poor microsomal stability was observed with these analogs (Table 4), suggesting that conformationally constrained analogs remain good substrates [45] for CYP3A4, as are many peptidomimetics, including PF74. A notable exception was analog **15** (t_1/2_ = 27 min) which contains a OH group at the C5 position of the indole ring.

### 3.5. Molecular Modeling

To analyze the observed SAR for pyridine and imidazole based chemotype 2 and chemotype 3, molecular modeling of key ligands was performed on the crystal structure of PF74 bound to native HIV-1 capsid protein (PDB code: 4XFZ [17]). Both (*S*)-configured-pyridine and -imidazole based analogs **9** and **20** were predicted to occupy the PF74 binding pocket, superimpose well with PF74, and interact with similar amino acid residues of the HIV-1 capsid protein (Figure 4A). The inhibitory activities of compounds **9** (EC_50_: 2.6 μM) and **20** (EC_50_: 3.7 μM) can be attributed to their backbone resemblance to, and common key interactions with PF74 (EC_50_: 0.7 μM), including (i) H-bonding between indole NH of compounds **9**, **20** and PF74 with Q63; (ii) cation-π interaction of the indole ring of the three compounds with protonated K70; (iii) H-bonding between the indole acetic acid carbonyl group and K70; and (iv) H-bonding between NH and carbonyl group of phenylalanine moiety of PF74 and terminal amide group of N57. The absence of phenylalanine carbonyl-O4 and N57 hydrogen-bonding in compounds **9** and **20** is likely to be partially compensated for by the bonding between the nitrogen atom of pyridine and imidazole rings and terminal amide of N57 (Figure 4A). A complete loss of potency was observed in the case of (*R*)-configured-pyridine based analog **4** (EC_50_ μM: >20 for **4** vs. 2.6 for its (*S*)-configured analog **9**). This is presumably because the benzyl group of compound **4** (circled) is expected to point away from the PF74-cavity, resulting in loss of hydrophobic interactions with M66 and other hydrophobic amino acid residues (Figure 4B). That, in turn, may lead to loss of the key H-bonding interaction between pyridine ring nitrogen atom and N57 (Figure 4B). Halogen-bonding was evident in current SAR as compound **13** showed a four-fold increase in potency compared to its non-halogen analog **9** (EC_50_ μM: 0.79 for **13** vs. 2.6 for **9**). In addition, the 5-hydroxy substitution on the indole ring of the pyridine analog **15** further enhanced antiviral potency 12- and 2-fold in comparison with compounds **9**, **13**, and PF74 (EC_50_ μM: 0.31 for **15** vs. 0.7 for PF74 vs. 0.79 for **13** vs. 2.6 for **9**). This increased potency can be attributed to the potential H-bonding by the OH of compound **15** to residue K182 on the adjacent CA_CTD_ (Figure 4C).

## 4. Discussion

PF74 is a phenylalanine-derived peptidomimetic with a well-defined binding mode to CA and sub-micromolar antiviral activity. Although mechanistically unique and synthetically highly accessible, PF74 is far less potent than GS-6207 and exceedingly unstable toward CYP-mediated oxidative metabolism. The three series of analogs (chemotypes 1-3, Figure 2) designed and synthesized in this study feature different forms of conformational restrictions of the PF74 backbone to improve antiviral potency and increased metabolic stability. Notably, both modifications (cyclization patterns A and B, Figure 2) involved altering the aniline *N*-methyl of PF74, which forms a key van der Waals interaction with the N53 required for the potency of the parent PF74 as per previously reported SAR [20]. Consistent with this SAR observation, modification A yielded compounds (chemotype 1) without appreciable antiviral activity or CA-stabilizing/destabilizing effect (Table 1). By contrast, modification B (Figure 2) is apparently tolerated as the resulting pyridine (chemotype 2) and imidazole (chemotype 3) series both contain compounds with significant antiviral activity and strong CA-stabilizing effect (Table 2 and Table 3), suggesting that the loss of the van der Waals interaction can be compensated for with modification B (chemotype 2 and chemotype 3) but not modification A (chemotype 1). As for modification B, while the SAR of the imidazole series (Table 3) was largely flat with the exception of the inactive analog **22,** the pyridine series displayed prominent SAR trends where both the antiviral potency and the CA-stabilizing effect were highly dependent on the *S*-configured phenylalanine moiety as well as the indole ring (Table 2).

Another important aim of the current work is to identify and avoid structural features with potential metabolic liability to improve metabolic stability. As mentioned earlier, the aniline *N*-methyl group of PF74 is highly susceptible toward oxidative metabolism. However, designing out this *N*-methyl group using a pyridine (**9**, **14**, **16**–**17**, Table 4) or imidazole (**20**, **22**) ring did not lead to significantly improved metabolic stability (Table 4). In the meantime, we have previously shown that replacing the electron-rich and easily oxidizable indole ring with less electron-rich ring systems could largely mitigate the metabolic liability [22,23]. Intriguingly, when an electron-donating hydroxyl group was introduced onto the C5 position of the indole ring, the resulting analog **15** showed drastically improved metabolic stability, despite being more electron-rich. This is presumably because the major oxidative metabolic pathway of the indole ring features a hydroxylation at the C5 position [46] as exemplified by the well-known conversion of tryptophan to serotonin [47]. Consistent with this metabolic pathway, **15** was also observed as a stable metabolite of **14**, **16** and **17**. The proposed metabolic pathways are described in Figure 5. The conversion of **16** to **15** likely entails a C5 hydroxylation and a de-ethylation at the indole N via α-carbon hydroxylation, followed by the collapse of the hemiaminal in **16b**. A similar hydroxylation of **17** could generate **14**, which was indeed observed as a metabolite of **17**. The de-methylation of the OMe group in **14** is likely via a hemiacetal intermediate **14a** produced by the α-carbon hydroxylation of the OMe. It is noteworthy that CYP-mediated de-alkylation is well-known [48].

## 5. Conclusions

To identify novel small molecules targeting the PF74 binding site of HIV-1 CA, we designed, synthesized and characterized three chemotypes featuring different conformational constraints. Through subsequent SAR, we identified analog 15 which displayed potent antiviral activity against HIV-1 (EC_50_ = 0.31 μM), strong CA-stabilizing effect (ΔTm = 8.7 °C), and drastically improved metabolic stability (t_1/2_ = 27 min). The observed metabolic pathways also suggest that blocking the C5 position of the indole ring may represent an important approach to metabolically stable PF74 analogs.

## Data Availability

The data presented in this study are Appendix A.

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
