# Peer review of "Design, Synthesis and Characterization of HIV-1 CA-Targeting Small Molecules: Conformational Restriction of PF74"

_viruses, 2021, doi:10.3390/v13030479_

Round 1

Reviewer 1 Report

In this work, the authors present a series of PF74-derived small molecules that intends to target HIV-1 CA. Their findings show that a modified series, which replaces the phenylalanine carboxamide moiety with pyridine or imidazole, offers promising results. The results indicate strong inhibition of HIV-1, strong stabilization of the CA hexamer and greatly enhanced metabolic stability. In general the manuscript is well written and its worthy of publication. However a few details should be addressed before the manuscript is published:

Review comments

L22: “Ligand binding mode?” – should this be “modes” or is there one binding mode? Similarly, “diversify ligand chemical class” is ambiguous as a non-expert. Is there one ‘chemical’ class of (singular) ligand? Minor point. 

L29: “… can provide viable hits.” Is this formal language? For a non-expert, could “hits” be clarified in this context? Minor point.

L56: “Structurally, CA monomer contains…” this is a grammatical error and should be amended. “Structurally, the CA monomer contains…” or “Structurally, CA monomers contain…” Minor point.

L62-72: Please list references for these statements, particularly about PF74 interactions and the CA-targeting chemotypes – even if redundantly reference 17 or 18. Minor point.

L86-87: Statement is unclear, was Q67H selected for via GS-6207 treatment in the clinical trial? If so, this should be made more clear. Minor point.

L95-96: Is it ‘typically’ true that molecular binding events involve entropy loss? In aqueous environments, drug/small molecule binding is often accompanied by expulsion of water from the binding site, which increases entropy and thus allows binding to evolve spontaneously. This language is dubious and should be clarified or supported with references.

L242: “… to an rmsd of 0.3 A…” grammatical error and should be amended. Either “… to a RMSD…” or “to a root-mean-square deviation of…”

L247: Why were VdW radii scaled by 0.8 for nonpolar atoms? If this is to allow flexibility in the modeling/subsequent screening, then it should be specified for reproducibility.

L248: Why was implicit solvent utilized for minimization? “To account for protein flexibility” is ambiguous. Further, how would the inclusion of both crystal water molecules and implicit solvent affect the protein structure during minimization?

Author Response

L22: “Ligand binding mode?” – should this be “modes” or is there one binding mode? Similarly, “diversify ligand chemical class” is ambiguous as a non-expert. Is there one ‘chemical’ class of (singular) ligand? Minor point. 

Response: Words “mode” and “class” are now changed to plural.

L29: “… can provide viable hits.” Is this formal language? For a non-expert, could “hits” be clarified in this context? Minor point.

Response: The word “hits” is indeed standard language in drug discovery and medicinal chemistry.

L56: “Structurally, CA monomer contains…” this is a grammatical error and should be amended. “Structurally, the CA monomer contains…” or “Structurally, CA monomers contain…” Minor point.

Response: Corrected to “Structurally, the CA monomer contains…”

L62-72: Please list references for these statements, particularly about PF74 interactions and the CA-targeting chemotypes – even if redundantly reference 17 or 18. Minor point.

Response: Thanks. Appropriate references are cited per your suggestion.

L86-87: Statement is unclear, was Q67H selected for via GS-6207 treatment in the clinical trial? If so, this should be made more clear. Minor point.

Response: Indeed, Q67H was selected for GS-6207 treatment in the clinical trial. We have further clarified this in the text.

L95-96: Is it ‘typically’ true that molecular binding events involve entropy loss? In aqueous environments, drug/small molecule binding is often accompanied by expulsion of water from the binding site, which increases entropy and thus allows binding to evolve spontaneously. This language is dubious and should be clarified or supported with references.

Response: Thanks for the comment. We have clarified the statement and cited a reference. We are discussing herein the conformational / configurational entropy contribution from the ligand: the binding event typically results in ligand restriction, and hence, a loss of the conformational / configurational entropy that needs to be compensated for.

L242: “… to an rmsd of 0.3 A…” grammatical error and should be amended. Either “… to a RMSD…” or “to a root-mean-square deviation of…”

Response: Changed to “RMSD”.

L247: Why were VdW radii scaled by 0.8 for nonpolar atoms? If this is to allow flexibility in the modeling/subsequent screening, then it should be specified for reproducibility.

Response: 0.8 is the default scaling factor of vdW for the ligand during regular molecular docking. The purpose is to soften the potential for nonpolar atoms to decrease penalties for close contacts in docking. We have clarified this by adding “to decrease penalties for close contacts” in the accompanying text.

L248: Why was implicit solvent utilized for minimization? “To account for protein flexibility” is ambiguous. Further, how would the inclusion of both crystal water molecules and implicit solvent affect the protein structure during minimization?

Response: Thanks for the comment. To describe more accurately what we did, we have changed the sentence in question to “all docked poses were subjected to post-docking minimization (PMD) to minimize a small number of poses within the field of the receptor to produce better poses”. PMD is also default setting in molecular docking.

Reviewer 2 Report

Study which highlights the design, synthesis and characterization PF74 related compounds based on the limitations of existing capsid targeting compounds. The primary aim is to generate a compound based on PF74 with increased metabolic stability and anti-HIV activity. They appear to achieve this with compound 15 that reaches IC50s of 300nM activity and importantly increased metabolic stability.

Strengths:

-Novel identification of a PF74 compound with nM activity and increased metabolic stability.

-Triaging of many other compounds that either show limited activity or similarly low metabolic stability to PF74.

Weaknesses:

-Primarily written in a format that is more suited to a chemistry journal submission. As I virologist, I did find it of interest, but it was at times very dense in format. The discussion for instance could do with a re-edit, with a greater focus on the overall impact of future impact of compound 15. That and making it more accessible to the audience of a virology focussed journal.

-Results were presented in a similar vain. Some primary data like does response curves for the final lead would have been nice to see. Instead it is buried in tables amongst many compounds with limited activity. 

-Whilst it was clear a lead had been found, it was less clear in the context of how consistent this result was and the statistics that would support a. significantly greater result than the other compounds.

-Whilst the inhibition in cell lines was great, it would have been great to see preliminary data for potency and toxicity in lead 15 in a primary T cell for instance. Often compounds work great in HeLa lines, but have significant toxicities in primary cells. 

-In the abstract I would move the "potent" in the description. 300nM is great, but many HIV inhibitions can reach very low nM or pM activities, so I would be careful in overstating observations here.

-On page 9 section 3.2, there is the mention of "significantly" inhibited HIV. Are there statistics to be aligned with this statement? At low uM concentrations, it does inhibit, but I would not term this level at significant levels of HIV inhibition. 

-On page 7 HIV is listed as a HIV strain. As one is only used, simple state HIV NL43.

-Page 6 lists "Biology Cells". Simply state cells or cell lines here. 

Author Response

Weaknesses:

-Primarily written in a format that is more suited to a chemistry journal submission. As I virologist, I did find it of interest, but it was at times very dense in format. The discussion for instance could do with a re-edit, with a greater focus on the overall impact of future impact of compound 15. That and making it more accessible to the audience of a virology focussed journal.

Response: Thanks for the feedback. We recognize that the work could be dense for pure virologists, but feel that the antiviral medicinal chemistry described herein fits the ‘capsid-targeting antivirals” theme of this special issue.

-Results were presented in a similar vain. Some primary data like does response curves for the final lead would have been nice to see. Instead it is buried in tables amongst many compounds with limited activity. 

Response: Again the point is well taken, but tabular data presentation is standard in antiviral medicinal chemistry. To address the reviewer’s suggestion, we have added shading to the entry for compound 15 (Table 2) to make it stand out. We also added the dose-response curve for hit compound 15 as the new Figure 3 in the text.

-Whilst it was clear a lead had been found, it was less clear in the context of how consistent this result was and the statistics that would support a. significantly greater result than the other compounds.

Response: Data are reported with a standard deviation from at least two independent experiments, as stated in table footnotes. Compound 15 was actually tested in four independent experiments (4 biological replicates with 2-3 technical replicates each), as it was a hit compound. We performed a standard two sample t-test to determine the significance of the EC50 value of hit compound 15 compared to the EC50 value of the parent PF74 compound. The p-value is 0.000758 with a T-score = 7.29, which demonstrate that difference between the average of these populations is large enough to be statistically significant. This information has been added to the text.

-Whilst the inhibition in cell lines was great, it would have been great to see preliminary data for potency and toxicity in lead 15 in a primary T cell for instance. Often compounds work great in HeLa lines, but have significant toxicities in primary cells. 

Response: This is an important point. We agree that it is important to test hit compounds in primary cells, which are more relevant in a biological context. As we reach advanced stages of the hit compound optimization process, we will assess potency of lead compounds in PBMCs and cytotoxicity in PBMCs and in primary human cardiomyocytes, lymphocytes, and hepatocytes.

-In the abstract I would move the "potent" in the description. 300nM is great, but many HIV inhibitions can reach very low nM or pM activities, so I would be careful in overstating observations here.

Response: The word “potently” is now changed to “effectively”.

-On page 9 section 3.2, there is the mention of "significantly" inhibited HIV. Are there statistics to be aligned with this statement? At low uM concentrations, it does inhibit, but I would not term this level at significant levels of HIV inhibition. 

Response: The word “significantly” is now changed to “considerably”.

-On page 7 HIV is listed as a HIV strain. As one is only used, simple state HIV NL43.

Response: We have updated the text as suggested.

-Page 6 lists "Biology Cells". Simply state cells or cell lines here. 

Response: We have updated the text as suggested.